# Smaller, Weaker, Yet Better: Training LLM Reasoners via Compute-Optimal Sampling

**Hritik Bansal**[1,2], **Arian Hosseini**[1,3], **Rishabh Agarwal**[1,3], **Vinh Q. Tran**[1], **Mehran Kazemi**[1]
[1] Google DeepMind, [2] UCLA, [3] Mila

## Abstract

Training on high-quality synthetic data from strong language models (LMs) is a common strategy to improve the reasoning performance of LMs.In this work, we revisit whether this strategy is compute-optimal under a fixed inference budget (e.g., FLOPs). To do so, we investigate the trade-offs between generating synthetic data using a stronger but more expensive (SE) model versus a weaker but cheaper (WC) model. We evaluate the generated data across three key metrics: coverage, diversity, and false positive rate, and show that the data from WC models may have higher coverage and diversity, but also exhibit higher false positive rates. We then finetune LMs on data from SE and WC models in different settings: knowledge distillation, self-improvement, and a novel weak-to-strong improvement setup where a weaker LM teaches reasoning to a stronger LM. Our findings reveal that models finetuned on WC-generated data consistently outperform those trained on SE-generated data across multiple benchmarks and multiple choices of WC and SE models. These results challenge the prevailing practice of relying on SE models for synthetic data generation, suggesting that WC may be the compute-optimal approach for training advanced LM reasoners.

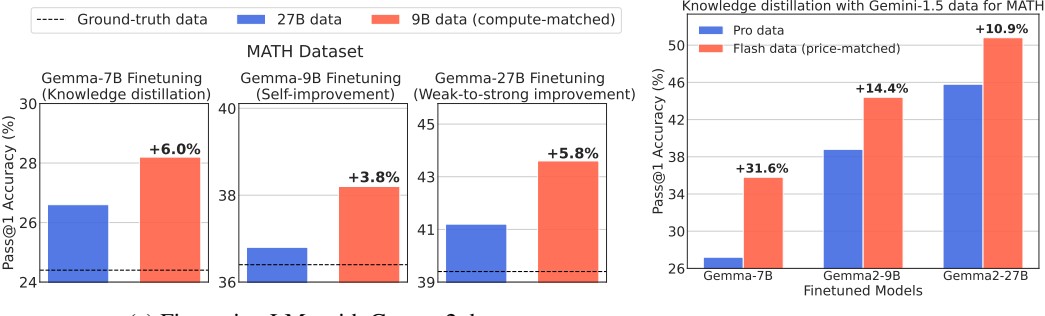

(a) Finetuning LMs with Gemma2 data.

(b) Finetuning LMs with Gemini 1.5 data.

Figure 1: **Summary of the results.** (a) We finetune Gemma-7B, Gemma2-9B, and Gemma2-27B on the synthetic data collected from a stronger but more expensive LM (Gemma2-27B) and a weaker but cheaper LM (Gemma2-9B) in a compute-matched setup for the MATH dataset. We find that training with Gemma2-9B data is a more compute-optimal setting across diverse finetuning paradigms – knowledge distillation, self-improvement, and weak-to-strong improvement (i.e. using a weaker model to improve a stronger model). (b) We finetune Gemma models (7B/9B/27B) on synthetic data generated by the state-of-the-art LMs Gemini-1.5-Pro and Gemini-1.5-Flash in a price-matched setup. We find that finetuning with Flash-generated data consistently outperforms Pro-generated data.

38th Conference on Neural Information Processing Systems (NeurIPS 2024) Workshop on MATH-AI.

# 1   Introduction

Language models (LMs) have demonstrated impressive capabilities in reasoning tasks, but their success heavily relies on being trained on vast amounts of (problem, solution) pairs. Collecting this data from humans is a costly and time-consuming process. Recent studies have demonstrated the feasibility of synthetically generating this data using LMs themselves, offering a potentially more scalable and efficient approach to training data acquisition. One such widely-adopted approach is to sample multiple candidate solutions for a problem from an LM, filters them for final answer correctness, and finetune models on the correct solutions [55, 34]. Practitioners often sample solutions from strong LMs to ensure high quality [41, 32, 29, 47]. However, sampling from strong LMs is expensive which limits the number of solutions that can be generated for practical sampling budgets.

In this paper, we explore an alternative sampling approach. Given a fixed compute budget, we investigate sampling from a **weaker but cheaper (WC)** model as opposed to the commonly-used approach of sampling from a **stronger but more expensive (SE)** model. We start by comparing data from WC vs SE across three axes that play crucial roles in the utility of such synthetic data: 1- *coverage*, the number of unique problems that are solved, 2- *diversity*, the average number of unique solutions we obtain per problem, and 3- *false positive rate (FPR)*, the percentage of problems that arrive at the correct final answer but with a wrong reasoning. We find that since we can generate more samples from the WC model compared to the SE model under a fixed budget, the data from WC may exhibit higher coverage and diversity. However, due to the lower quality of the WC model, it may also have a higher FPR. As a particular example for the Gemma2 family [38, 39] on the MATH dataset [15], Gemma2-9B achieves $11\%$ higher coverage and $86\%$ higher diversity, but also with $7\%$ higher FPR compared to Gemma2-27B.

We then fine-tune models on data from SE and WC at a fixed compute budget (see Appendix Figure 3 for illustration). We consider diverse setups corresponding to three paradigms: 1) *knowledge distillation*, where a student LM learns from a teacher LM [18]; 2) *self-improvement*, where an LM learns from self-generated data [20]; and 3) a new paradigm we introduce called *Weak-to-Strong Improvement*, where a strong student LM improves using synthetic data from a weaker teacher LM. Using two (WC, SE) model pairs, one from the Gemma2 family and another from the Gemini-1.5 family [31], we show on multiple benchmarks that training on WC-generated data consistently outperforms training on SE-generated data under the three setups, with relative gains of up to $31.6\%$ percent (see Figure 1 for a summary of the results). Our results indicate that it is more compute-optimal to sample from a WC model as opposed to the common-practice of sampling from a SE model. With the performance gap between small and large LMs getting narrower over time (especially at larger scales), our results establish a solid foundation for training the next generation of LM reasoners.

# 2   Compute-Matched Sampling and Training

We present the background on synthetic data generation and supervised finetuning in Appendix §A. To generate synthetic solutions for the problems in the dataset, one can leverage different models as data generators. Specifically, at a fixed sampling budget (FLOPs), one can generate more samples from a weaker but cheaper (WC) model or fewer samples from a stronger but more expensive (SE) model. Given a WC model with $P_{WC}$ parameters and SE with $P_{SE}$ parameters, we compute the sampling ratio at a fix budget for the two models, focusing on decoder-only transformer models [43]. Following [23], we note that the FLOPs per inference token is $2P$, for a model with $P$ parameters. As a result, the FLOPs for $T$ inference tokens is $2PT$. Further, we assume that generating each solution requires an average of $W$ inference tokens for both models. Let $S_{WC}$ and $S_{SE}$ represent the number of samples we generate per question for the two models. The total cost of generating samples for the dataset $\mathcal{D}$ will then be $Cost_{WC} = n \times S_{WC} \times W \times (2P_{WC})$ and $Cost_{SE} = n \times S_{SE} \times W \times (2P_{SE})$ for the cheap and expensive models, respectively. At a fixed sampling budget, we have:

$$n \times S_{WC} \times W \times (2P_{WC}) = n \times S_{SE} \times W \times (2P_{SE}) \quad \Rightarrow \quad \boxed{S_{WC} = \frac{P_{SE}}{P_{WC}} S_{SE}} \quad (1)$$

Equation 1 indicates that at a fixed sampling budget, for each question we can generate $P_{SE}/P_{WC}$ more samples from WC; the ratio scales linearly with the model parameters ratio. Sampling more solutions from WC may increase the likelihood of correctly solving a larger subset of the problems (high coverage) and obtaining more correct solutions per question (high diversity).

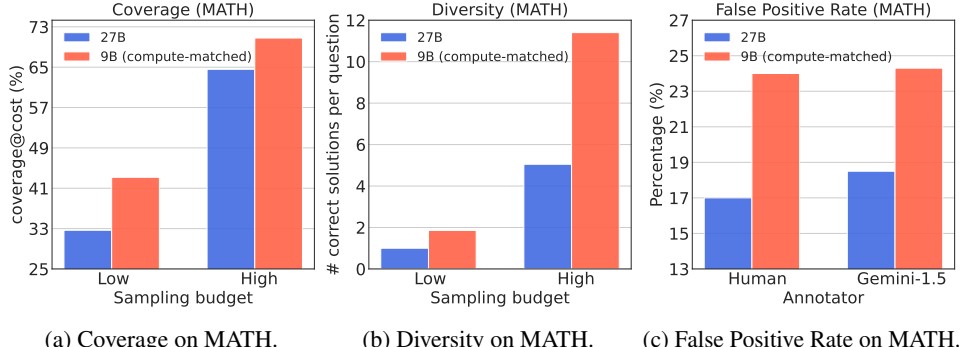

(a) Coverage on MATH.  (b) Diversity on MATH.  (c) False Positive Rate on MATH.

Figure 2: **Synthetic data analysis for MATH dataset.** The (a) coverage, (b) diversity, and (c) false positive rates for Gemma2-27B and Gemma2-9B on the MATH dataset, at two sampling budgets.

## 3 Experiments and Results

We experiment with MATH and GSM-8K datasets, and collect synthetic data from Gemma2-9B (WC) and Gemma2-27B (SE) models (see Appendix C for other experimental details).

### 3.1 Synthetic Data Analysis

**Coverage:** Here, we aim to understand the pros and cons of generating solutions from the WC and SE models at a fixed sampling budget. We present the coverage, diversity, and FPR for the MATH at the low and high sampling budgets in Figure 2. The results for GSM-8K are presented in the Appendix – Figure 15. We find that in terms of coverage, the data from Gemma2-9B (WC) outperforms Gemma2-27B (SE) by $11\%$ and $6\%$ at the low and high sampling budgets, respectively, for the MATH dataset, and $8\%$ and $1\%$ for GSM-8K. This highlights that the higher number of samples for the WC model aids in solving more unique problems for both the reasoning datasets. We provide the coverage trends for diverse sampling budgets in Appendix J. Further, we provide a qualitative example that gets solved by repeated sampling from Gemma2-9B but remains unsolved by Gemma2-27B at the fixed high sampling budget (Table 5).

**Diversity:** The diversity for the data from Gemma2-9B is higher than Gemma2-27B by $86\%$ and $125\%$ at the low and high sampling budgets for the MATH dataset, and $134\%$ and $158\%$ at for the GSM-8K dataset. This implies that many unique reasoning chains in the synthetic data from the WC model lead to the correct solutions. We also observe that the absolute diversity scores are lower for MATH compared to GSM-8K at high sampling budget, indicating that models generate fewer correct solutions for the more challenging datasets when using repeated sampling.

**FPR:** Since we utilize the final answer correctness for filtering the synthetic data, it does not remove the solutions with incorrect intermediate reasoning steps. Our human evaluations suggest that the FPR for the WC-generated solutions is $7\%$ and $2\%$ higher than SE-generated solutions on the MATH and GSM-8K, respectively. The trends from the automatic evaluation are similar to that of human evaluation. Due to the differences in the difficulty of the problems, we note that the absolute FPRs are much lower for the GSM-8K dataset as compared to the MATH dataset. We also note that the automatic verification of the reasoning steps can also have errors and is still an open problem [28].

Given the mixed signals of high coverage and diversity coupled with a high FPR, it remains unclear whether it is compute-optimal to sample from the WC model or the SE model for training strong reasoners. We study this in the next section.

### 3.2 Compute-Optimality Results for Training

**Student-LM Finetuning.** Here, we aim to understand the merit of WC data compared to SE data for distillation to a student LM (Gemma 7B in our experiment), given a fixed sampling budget. The results presented in Figure 1a show that the Gemma-7B finetuned with the synthetic data from WC consistently outperforms the one finetuned on data from SC. We observe relative gains of $6\%$ for

the MATH dataset. Contrary to the common belief of stronger models being better for knowledge distillation, our results indicate that finetuning on data from WC is more compute-optimal than SE.

**WC-LM Finetuning.** Prior work [34] has shown that finetuning a WC model through self-generated data lags behind distillation from SE data. Here, we revisit this setup under the fixed sampling budget and compare WC models finetuned with WC data vs SE data. Specifically, we compare the performance of Gemma2-9B finetuned with the WC data (i.e. self-generated data) and SE data (i.e. data from Gemma2-27B). In Figure 1a, we observe that the self-generated data (WC data) improves over knowledge distillation from a strong model (SE data), achieving relative gains of $3.8\%$ for the MATH dataset. Interestingly, our findings suggest that training a WC model on its own synthetic data is more compute-optimal than distillation from a stronger model.

**SE-LM finetuning.** It is commonly believed that to improve a SE model, we either need synthetic data from the SE model itself or from an even stronger (and perhaps more expensive) model than SE. Here, we test an alternative novel approach, which we term *weak-to-strong improvement (W2S-I)*, to understand whether the synthetic data from the WC model can improve the SE model. We present the results for finetuning Gemma2-27B with the Gemma2-9B generated data and self-generated data (Figure 1a). Surprisingly, we observe that the model finetuned with the WC data outperforms the SE data, achieving relative gains of $5.8\%$ for the MATH dataset. Contrary to the common belief of self-generated data or data from a stronger model being better, our empirical findings show that training a model in a W2S-I setup from a WC data may be more compute-optimal than training it in a self-improvement setup on its own data. This result also establishes a new paradigm for improving frontier models in a compute-efficient way, by generating synthetic data from much smaller models.

We present the results for more sampling budgets and datasets in Appendix Figure 4 and 5. We present the generalization and ablation results in Appendix §D and §E, respectively.

# 4 Scaling to state-of-the-art language models

We scale our method to sampling data from Gemini-1.5-Pro and Gemini-1.5-Flash. As the model sizes are not publicly available, we utilize the ratio between their *pricing per output token* as a proxy to perform compute-matched sampling. As of August 2024, we note that the price per million output tokens is \$10.5 and \$0.3 for Gemini-1.5-Pro and Gemini-1.5-Flash, respectively. Hence, we sample 1 and 35 solutions per problem from 1.5-Pro and 1.5-Flash, respectively, for MATH dataset.

We perform knowledge distillation on the Gemma-7B, Gemma2-9B, and Gemma2-27B LMs with the synthetic data from the Pro (SE) and Flash (WC) models. We present the results in Figure 1b. Interestingly, we find that finetuning with the WC data outperforms the SE data, achieving relative gains of $31.6\%$, $14.4\%$, and $10.9\%$ for Gemma-7B, Gemma2-9B, and Gemma2-27B, respectively. This can be attributed to the difference in the coverage of the models at the fixed sampling budget, which is $61.1\%$ and $81\%$ for 1.5-Pro and 1.5-Flash, respectively.

Further, we investigate training the LMs with the WC data that is less expensive than collecting 1 solution per problem from the SE model. Specifically, we create a dataset by sampling 5 solutions per problem from the Flash (WC) model, which is $7\times$ more economical than generating 1 solution from the Pro (SE) model, in terms of the price (\$). Upon training the LMs on the $0.15\times$ *cost* data regime, according to Figure 7 in the Appendix, we find that training on this data can also outperform training with SC data, achieving relative gains of $19.1\%$, $9.8\%$, and $5.7\%$ for finetuning Gemma-7B, Gemma2-9B, and Gemma2-27B, respectively. This can be attributed to higher coverage of the weaker model ($69\%$), even in the more economical scenario, in comparison to the stronger model ($61.1\%$).

# 5 Conclusion

In this work, we provide a framework for compute-optimal sampling from weak but cheap LM for reasoning tasks. We show that at a fixed sampling compute budget, repeated sampling from a smaller model can achieve higher coverage and diversity than from a strong but more expensive model. Furthermore, we find that finetuning LMs with data from the small LM can consistently outperform data from the large LM under the same compute budget. Our results can serve as a foundation for training LM reasoners, especially as the performance gap between small and large LMs continues to narrow over time.

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

# A Preliminaries

Let $\mathcal{D} = \{q_i, a_i\}_{i=1}^{i=n}$ be a training dataset of size $n$ with reasoning questions $q_i$ and final answers (aka labels) $a_i$. A successful approach to leverage such data to improve models for reasoning is as follows. We sample multiple solutions for each $q_i$ at a non-zero temperature and create the synthetic data $\mathcal{D}_G = \{q_i, \{(\hat{r}_{ij}, \hat{a}_{ij})_{j=1}^{j=k}\}\}$, where $k$ is the number of samples, $\hat{r}_{ij}$ is the $j$-th reasoning chain (i.e. solution) generated by the model for $q_i$, and $\hat{a}_{ij}$ is the model's final answer for $q_i$ in the $j$-th sample. Then, we filter the incorrect solutions by comparing $\hat{a}_{ij}$ to $a_i$ and removing the solutions whose final answer do not match that of the gold answer[1]. Finally, we supervise finetune a model on the remaining data $\tilde{D}_G$. This approach was first proposed in [55] and was then extended in multiple works including [54, 34].

For a dataset $\mathcal{D}_G$, we compute $coverage@k$ (aka $pass@k$) [9] as $\mathbb{E}_{\mathcal{D}_G}\left[1 - \binom{M-c}{k}/\binom{M}{k}\right]$ where $c$ is the number of solutions, out of $M$, with correct answers and $\mathbb{E}_{\mathcal{D}_G}[.]$ denotes the expectation over the problems and solutions in the generated dataset. Conceptually, $coverage@k$ measures the fraction of *unique* questions that have at least one correct solution, assuming that we sample $k$ solutions per question from the model. We also define $diversity@k$ as the average number of unique correct solutions we obtain per question when we sample $k$ solutions per question. Finally, we define *false positive rate (FPR)* as the percentage of solutions in $\tilde{D}_G$ where the reasoning is incorrect, despite the final answer being correct.

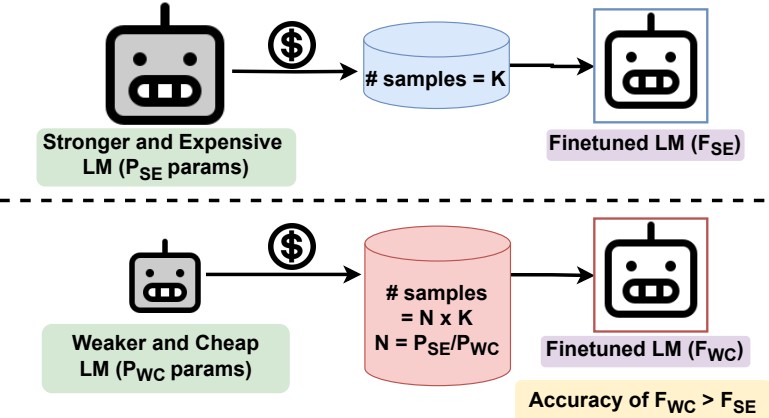

Figure 3: **Illustration of the approach.** Given a fixed sampling budget, one can either generate fewer samples from a stronger but more expensive (SE) model or more samples from a weaker but cheaper (WC) model. The latter may lead to solving a wider range of problems and also more correct solutions per question. We compare the utility of these two synthetically generated datasets for training LM reasoners in various supervised finetuning setups and show that training with the data from WC consistently outperforms training on data from SE.

| Data ($\downarrow$) / Finetuning setup ($\rightarrow$) | Student-LM | WC-LM | SE-LM |
|---|---|---|---|
| **WC (Compute-matched)** | Knowledge distillation | Self-improvement | Weak-to-strong improvement |
| **SE** | Knowledge distillation | Knowledge distillation | Self-improvement |

Table 1: **Summary of the supervised finetuning setups.** We finetuned the language models under three setups: (a) Student LM, (b) Weak-Cheap (WC) LM, and (c) Strong-Expensive (SE) LM. For each setup, we employed different finetuning paradigms based on the source of the synthetic data. For example, training a separate student LM with data from both WC and SE models falls under the knowledge distillation paradigm. In contrast, training a WC model with its own samples is self-improvement. Finally, we also introduce a new paradigm, weak-to-strong improvement, where the samples from the WC model is used to improve the reasoning capabilities of the SE model at the fixed compute budget.

---

[1]While it is possible to use more sophisticated approaches for filtering (e.g., process-based or outcome-based reward model [42]), in this work we focus on final answer correctness for filtering as it has shown to be strong.

# B  Finetuning setups

- **Student-LM finetuning**: Conventionally, the supervised finetuning data for training student LM is acquired from SE models to ensure high-quality [41]. However, we aim to understand whether WC models can replace SE models for distillation at the fixed sampling budget. To do so, we finetune a student LM separate from the WC and SE models on the WC and SE data, which corresponds to distillation in both the cases.

- **WC-LM finetuning**: Prior work [34] has shown that finetuning a WC model through self-generated data lags behind distillation from SE data. However, their setup spends a higher sampling budget (FLOPs) on collecting data from the SE model than collecting SI data from the WC model. In this work, we revisit this finetuning setup under the fixed sampling budget and finetune the WC model on the WC and SE data at a fixed budget for both. Note that training the WC model on its own data corresponds to self-improvement whereas training WC on the data from SE corresponds to distillation. Hence, this setup compares self-improvement on WC data with distillation from SE data.

- **SE-LM finetuning**: It is commonly believed that to improve a SE model, we either need synthetic data from the SE model itself or from an even stronger (and perhaps more expensive) model than SE. Here, we test an alternative approach to understand whether the synthetic data from the WC model can improve the SE model. To this end, we finetune the SE model on the WC and SE data. Training SE on data from WC corresponds to W2S-I and training SE on data from SE corresponds to self-improvement. Overall, this setup compares W2S-I by WC data with self-improvement by SE data.

A summary of the three setups and the finetuning paradigms that each case corresponds to can be found in Table 1.

# C  Experimental Setup

**Datasets:** We utilize MATH [15] and GSM-8K [10] as the reasoning datasets due to their wide adoption for mathematical problem solving. Specifically, MATH consists of competition level problems with various levels of difficulty (Level 1-5), and GSM-8K comprises of grade school level math problems. Each dataset contains 7500 math problems in their training split. We evaluate the models on 500 problems from the MATH test split [28] and 1319 problems from the GSM-8K test split. Further, we use 500 problems from the MATH test split and 500 problems from GSM-8K as the validation dataset. We also use the Functional MATH dataset [37] for a transfer study. Further, we present the results for a coding task in Appendix H.

**Data Generation:** We use Gemma2 models for synthetic data generation, with pretrained Gemma2-9B and Gemma2-27B acting as the WC and SE models respectively. We generate the solutions for the problems in the MATH using a 4-shot prompt and for GSM-8K using an 8-shot prompt. Since the 9B model is roughly 3 times smaller than the 27B model, at a fixed sampling compute budget we can sample $3\times$ more sample solutions per problem for Gemma2-9B. For our experiments, we consider two sampling budgets: a *low budget*, where we generate 1 and 3 candidate solutions per problem from Gemma2-27B and Gemma2-9B, respectively, and a *high budget*, where we generate 10 and 30 candidate solutions per problem. Further, we study the transfer of the reasoning capabilities for the models trained on MATH at the high sampling budget on the Functional MATH dataset.

**Model Finetuning:** We summarize the details for our finetuning setups in the Table 1. In the Student-LM finetuning setup, we finetune the Gemma-7B model [38] on data from Gemma2-9B (WC) and Gemma2-27B (SE). In addition, we use Gemma2-9B and Gemma2-27B for the WC-LM and SE-LM finetuning setups, respectively. Further, we train the LMs across different setups with the human-written solutions as a ground-truth baseline. We provide the finetuning details in Appendix M.

**Synthetic Data Evaluation:** To assess the quality of the synthetic data from the SE and WC models, we measure the false positive rate, as well as *coverage* and *diversity* at a fixed cost. From Equation 1, we know that sampling one solution from SE takes the same FLOPs as sampling $P_{SE}/P_{WC}$ solutions from WC. Therefore, we compare $coverage@k$ for SE to $coverage@(\frac{P_{SE}}{P_{WC}}k)$ for WC to allow a similar budget to both models. Specifically, we compare $coverage@k$ and $coverage@3k$ for our SE and WC models. Similarly we compare $diversity@k$ and $diversity@3k$ for our SE and WC

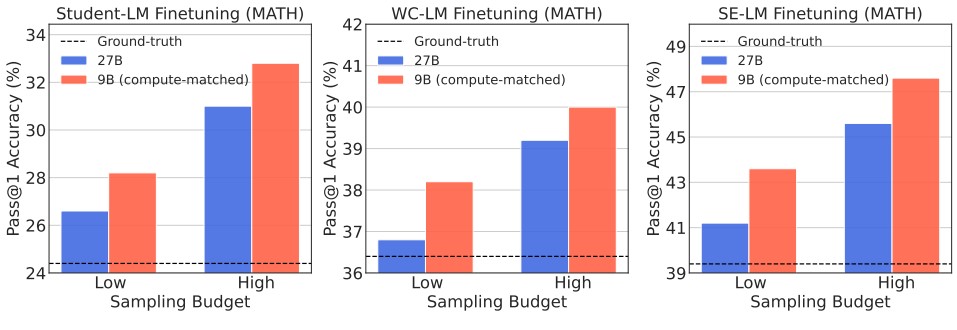

(a) Finetuning Gemma-7B.    (b) Finetuning Gemma2-9B.    (c) Finetuning Gemma2-27B.

Figure 4: **Supervised-finetuning results (MATH).** The results for finetuning various LMs on the MATH synthetic data from the WC (Gemma2-9B) and SE (Gemma2-27B) models, at a fixed sampling budget. We observe that training with the samples from the WC model consistently outperforms training with SE data.

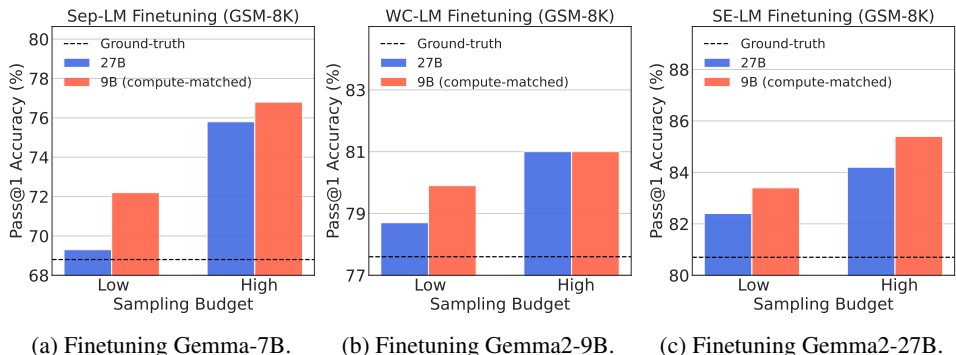

(a) Finetuning Gemma-7B.    (b) Finetuning Gemma2-9B.    (c) Finetuning Gemma2-27B.

Figure 5: **Supervised-finetuning results (GSM-8K).** The results for finetuning various LMs on the GSM-8K synthetic data from the WC (Gemma2-9B) and SE (Gemma2-27B) models, at a fixed sampling budget. We observe that training with samples from the WC model leads to stronger reasoners than training with SE data.

models. Since FPR cannot be computed automatically, we compute it using two proxies: 1- a human evaluation on a subset of the data, where $50$ solutions from each model were selected randomly and rated for reasoning correctness by the authors, and 2- automatic evaluation where we sampled $500$ solutions and prompted Gemini-Pro-1.5 [31] to rate the correctness of the reasoning paths. To sample solutions, for the MATH dataset we selected uniformly from each diversity level. In our experiments, we find that the FPR estimates are close to each other for the human and automatic evaluation. We provide a few qualitative examples for the false positive instances in Appendix I.

**Evaluating Finetuned Models:** We use pass@1 accuracy to evaluate the performance of the finetuned LMs. Specifically, we generate a single solution for the problem (zero-shot) from the test split, using a sampling temperature of $0.0$ (greedy decoding) for the fine-tuned LM and measure the percentage of problems that where the final answer matches the golden final answer. We also report maj@k ($k = 1, 4, 8, 16$) for part of our experiments, where we generate $k$ solutions per problem at a sampling temperature of $0.7$ and select the final answer that appears most among the $k$ samples.

## D   Generalization

Here, we aim to study the transfer capabilities of the models trained with the WC and SE data. Specifically, we evaluate the models finetuned with the synthetic solutions for the MATH datasets at the high sampling budget on the Functional MATH dataset. The results in Figure 6 show that the Gemma-7B finetuned with the WC data consistently outperforms the SE data, where the relative gains range from $5.8\% - 6.5\%$ at different values of $k$. In addition, we observe that the Gemma2-9B

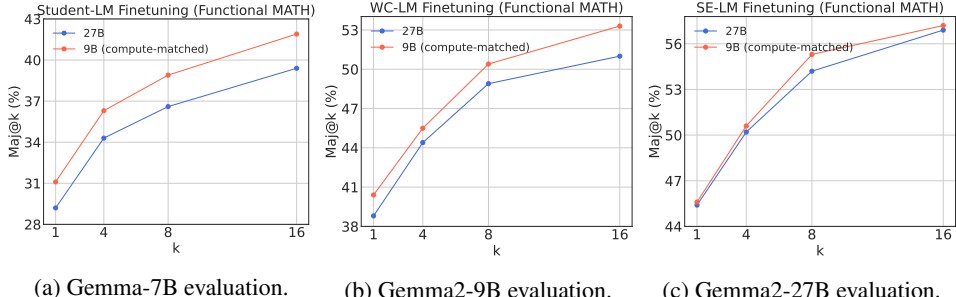

(a) Gemma-7B evaluation.  (b) Gemma2-9B evaluation.  (c) Gemma2-27B evaluation.

Figure 6: **Generalization Results (Functional MATH).** The performance of the models trained with the synthetic data from the MATH data at high sampling budget on the Functional MATH dataset. The results suggest that training with WC data enhances the generalization capabilities over the SE data, at a fixed sampling budget.

finetuned with the self-generated data outperforms knowledge distillation with the Gemma2-27B data achieving relative gains ranging from $2.5\% - 4.5\%$ at different values of $k$. Moreover, finetuning Gemma2-27B with WC data matches closely with the SE data, except for $k = 8$ where the gap is a relative gain of $2\%$. Our results highlight that finetuning the LMs with the WC data enhances the generalization capabilities over the SE data at the fixed sampling budget.

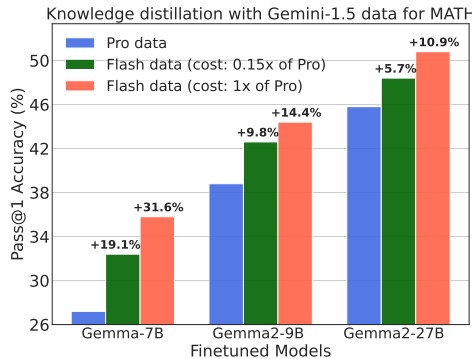

Figure 7: We finetune Gemma models (7B/9B/27B) on synthetic data generated by the state-of-the-art LMs Gemini-1.5-Pro and Gemini-1.5-Flash. We find that finetuning with Flash-generated data consistently outperforms Pro-generated data not only at the same sampling monetary cost as Gemini-1.5-Pro, but also at $\approx 0.15\times$ of the cost.

# E   Ablation Studies

**Impact of Dataset Size:** We study whether the benefits of the synthetic data from the WC model hold at different dataset sizes. We repeat our experiments for the MATH dataset at the high budget, but when only having access to $500$ training data (selected randomly from the training set). We present the results for the finetuned models in Figure 8. We observe that models trained with the WC data outperform those trained with the SE data, achieving relative gains of $12.93\%$, $11.4\%$, and $5.1\%$ for the three paradigms, respectively. This highlights the utility of generating more data from the WC model instead of the SE model in the low-problem regimes at the fixed sampling budget.

**Default vs Compute-Optimal Sampling from Cheap LMs:** We anticipate that the reason why data from SE models has been previously preferred over data from WC is because they have been tested in a setup where an equal number of samples have been generated from the two models (e.g., see [34]), as opposed to a compute-matched setup. To verify this, we generated 1 solution per problem (number-matched) from the WC model for the MATH and GSM-8K datasets and trained the models under the three fine-tuning setups on this generated data, after filtering for final answer correctness. We then compare the performance of the models trained with synthetic data, where

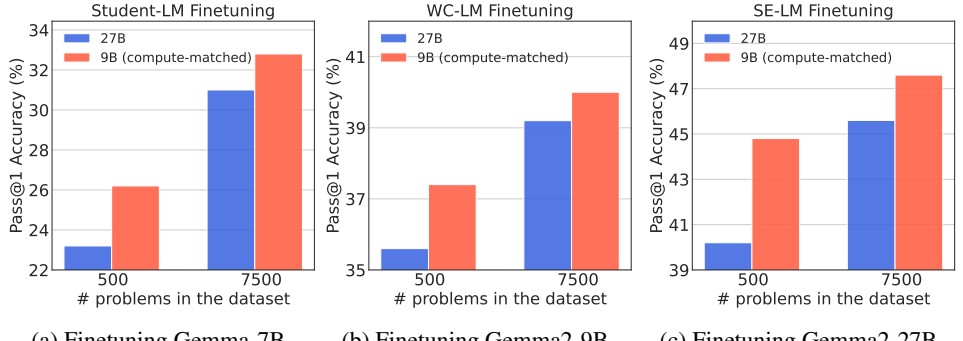

(a) Finetuning Gemma-7B.  (b) Finetuning Gemma2-9B.  (c) Finetuning Gemma2-27B.

Figure 8: **Impact of the dataset size.** The performance of finetuned LMs on the synthetic data from WC and SE models, at different sizes of the training set. Training with the WC data leads to better models than training with the SE data at both dataset sizes.

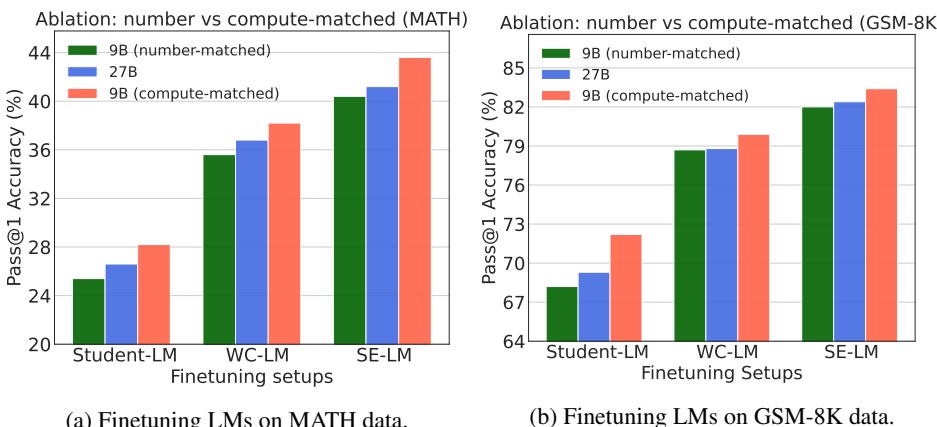

(a) Finetuning LMs on MATH data.  (b) Finetuning LMs on GSM-8K data.

Figure 9: **Comparison between number-matched sampling and compute-matched sampling from the WC model.** We report the results for finetuning diverse LMs with synthetic data from WC and SE model at the low sampling budget. Conventionally, practitioners would compare the performance of the models trained with WC data and SE data at the fixed *number* of samples from both models. However, we observe larger gains using the samples from WC model that acquired at the fixed *sampling* budget as that of SE model.

we generate 3 solutions per problem from the WC model, matched in sampling compute to the SE model. We present the results in Figure 9. We see that the models trained with the number-matched WC data are sub-optimal in comparison to the models trained with the compute-matched WC data, and lead to worse models compared to training with the SE data. This highlights that the future comparisons between synthetic data from weak and strong models should be made in the sampling compute-matched regime.

**Coverage and Diversity:** We aim to understand the role of coverage and diversity in enhancing the performance of models trained with WC-generated synthetic data. To this end, for the MATH dataset, we consider the original high-sampling (30 solutions per problem) WC dataset as a *(high coverage, high diversity)* dataset. We then construct a *(high coverage, low diversity)* version by only selecting one correct solution per question from our samples. This reduces the diversity of the original WC dataset from 11 to 1, while maintaining the coverage. We also create a *(low coverage, low diversity)* dataset where we generate just one solution per problem from the WC model and filter it for the correctness of the final answer. The coverage of this dataset ($27\%$) is lower than that of the WC dataset with 30 solutions per problem ($43\%$). We train models across the three finetuning setups on these sets and present the results in Figure 10. Our results indicate that across all setups, the high coverage and high diversity data is better than high coverage and low diversity, and high coverage and low diversity is better than low coverage and low diversity. This reveals that both the coverage and diversity play a critical role in training strong reasoners from the smaller LMs.

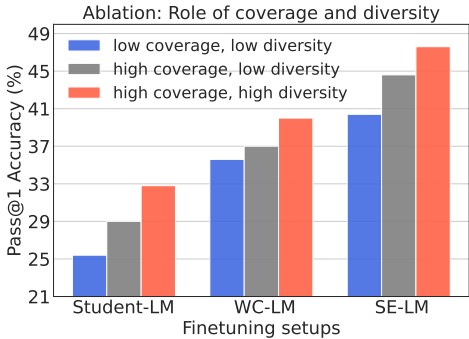

Figure 10: **Understanding the role of coverage and diversity for training strong reasoners with WC model.** We compare the performance of training the LMs with synthetic data acquired by collecting (a) 1 solution per problem (low diversity, low coverage), (b) 30 solutions per problem (high diversity, high coverage), and (c) 30 solutions per problem but keeping just one correct solution (high coverage, low diversity). We find that both high diversity and coverage are helpful for training strong reasoners.

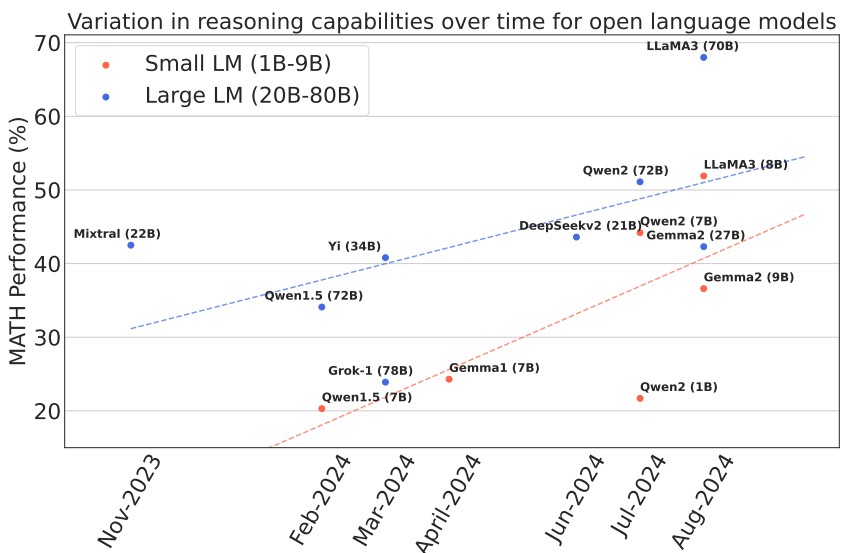

Figure 11: **Variation in the performance of open language models on the MATH dataset over time.** The fitted trendlines suggest that the quality of smaller LMs is improving more rapidly than that of larger LMs over time. This highlights that our findings on utilizing smaller LMs for training strong reasoners will become increasingly relevant in the future.

# F   A Future Perspective

We showed that for the current WC and SE models, training reasoners through sampling from WC models may be more compute-optimal. Here, we aim to discuss the relevance of these results for the future set of WC and SE models. To do so, we surveyed 17 LMs that pass the following criteria: 1- the model size is known and falls within [1B, 9B] or [20B, 80B] range, 2- the model is released in the past one year, 2- the technical report of the model reports results on the MATH dataset and the model is capable on it ($> 20\%$), 4- ranks high on the OpenLLM leaderboard under the pretrained models category [16]. This resulted in models from seven families including Gemma-2 [39], LLaMA-3 [11], Mixtral [22], Qwen [40, 48], Grok-1 [46], DeepSeek-v2 [33], and Yi [50]. We grouped these models into small LM (1B to 9B) and large LMs (20B to 80B). We then plotted in Figure 11 the model performances on the MATH dataset against their date of the publication release on arxiv and fitted trendlines to the data points representing the small and large LMs using the least squares method[2].

---

[2]We consider the number of active model parameters for mixture-of-experts LMs.

Our analysis reveals that, despite the variance, the trendline for the smaller LMs is steeper than that of the larger LMs. This indicates that the reasoning performance of the small LMs may be improving more rapidly over time compared to the larger LMs. The rapid rise in the performance of the small LMs can be attributed to factors such as the enhanced quality and scale of the pretraining data (e.g., LLaMA-3 employs 15T tokens), pruning and knowledge distillation [30]. With the performance gap between small and large LMs narrowing over time, we anticipate that our results will become even more relevant in the future.

# G    Related Work

**LMs for reasoning.** The ability to solve reasoning tasks has been a long standing goal of artificial intelligence [31, 1, 11, 40, 3, 2]. In this regard, LMs trained on the internet-scale data have achieved great success for math, code, and other reasoning tasks [27, 5, 24]. There have been several works that aim to enhance the reasoning capabilities of the LMs either via prompting [26, 44, 56, 25] or finetuning [53, 51]. In this work, we focus on finetuning the LMs with task-specific datasets to build strong reasoners. Specifically, our method closely aligns with the widely adopted STaR [55] where the synthetic data from the LMs are used to elicit strong reasoning capabilities.

**Finetuning LMs.** Within the finetuning paradigm, there have been several works that improve reasoning with synthetic data. Broadly, these works focus on knowledge distillation from a strong but expensive LM [45, 53] or self-improvement [12, 34]. While it is common to filter the synthetic data for the final answer correctness (akin to [55]), there are several works that aim to build task-specific verifiers to train strong reasoners [28, 45, 19, 52]. In this work, we explore the utility of the synthetic data from the weak but cheap LMs for training strong reasoners via knowledge distillation as well as self-improvement. However, we do not explore using model-based verifiers with the synthetic data for enhanced reasoning, and leave it as a future work.

Our weak-to-strong improvement paradigm, where a strong model is trained with the generations from the weak model, is related to several prior work [6, 8, 49] which study the ability of a strong LM to learn from the data generated by a weaker LM. However, the aim of these works is to recover the full capabilities of the strong model from weaker data, whereas we aim to enhance the strong model capabilities further. Additionally, our work studies compute-optimal sampling from weak and strong models, which is absent in previous work.

**Large and small LMs.** While training large LMs has led to significant advancements across various tasks, there has recently been a growing interest in developing capable small LMs [17, 21]. Specifically, a capable small LM is faster to run, and easier to serve to millions of users on the edge devices [13]. As a result, several recent works aim to understand the utility of the weak but cheaper LMs in comparison to the strong but expensive LMs for reasoning. Specifically, [7, 36, 35] show that the solve rate of the small LMs can increase significantly with repeated sampling. In addition, [14] demonstrate that repeated generations from smaller LMs can outperform the data generated by larger LMs at a fixed sampling computational budget during inference for coding tasks. In this work, we go beyond these works and show the utility of the synthetic data from the small LMs for training strong reasoners across a diverse set of supervised finetuning setups.

# H    Extending our results to coding tasks

Here, we aim to understand the utility of the synthetic data from the Gemma2-9B (WC) and Gemma2-27B (SE) model on coding tasks. To this end, we generate candidate solutions for the MBPP [4] dataset from WC and SE models at the low and high sampling budgets and finetune models in three setups on these data. We use the santizied version of MBPP[3] containing 427 problems overall; we used 3 problems for fewshot prompting (used for sampling from the models), 324 problems for synthetic training data generation, and 100 problems for validation. The candidate solutions are filtered by the unit tests that accompany each instance of the dataset. After finetuning, we evaluate the LMs on 164 problems from the HumanEval dataset [9].

We compare the coverage and diversity of the synthetic datasets in Figure 12 and observe that the coverage of the WC model is higher than SE at low data regime while it is similar to SE in the high

---

[3]https://huggingface.co/datasets/google-research-datasets/mbpp/viewer/sanitized

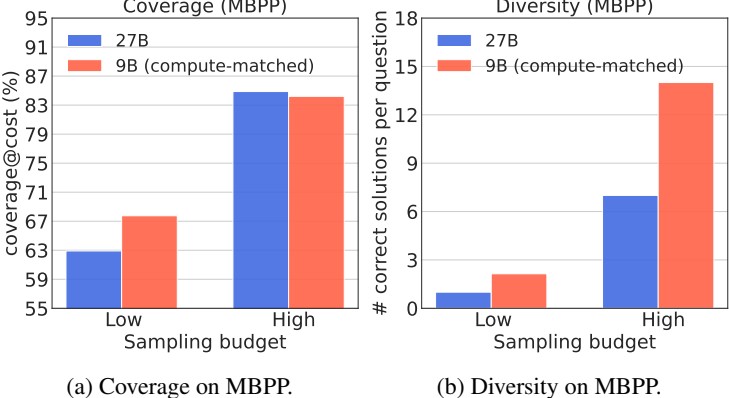

(a) Coverage on MBPP.      (b) Diversity on MBPP.

Figure 12: **Synthetic data analysis for MBPP dataset.** We present the (a) coverage, and (b) diversity for a subset of the santized MBPP dataset for Gemma2-27B and Gemma2-9B at two fixed sampling budgets.

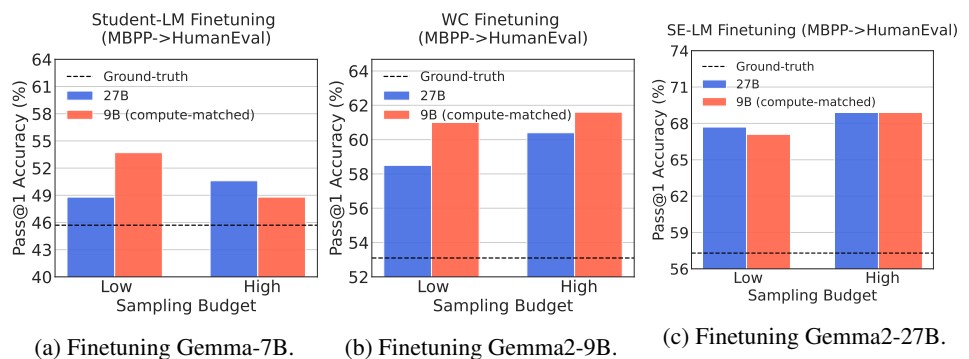

(a) Finetuning Gemma-7B.  (b) Finetuning Gemma2-9B.  (c) Finetuning Gemma2-27B.

Figure 13: **Supervised-finetuning with MBPP and evaluation on HumanEval.** We report the results for finetuning diverse language models on the MBPP synthetic data from the SE model (Gemma2-9B) and WC model (Gemma2-27B) at the fixed sampling budgets.

sampling budget regime. In addition, we find that the diversity of the WC model is more than that of the SE model for the low and high sampling budgets. Subsequently, we finetune Gemma-7B, Gemma2-9B, and Gemma2-27B models with the ground-truth and synthetic datasets and evaluate on HumanEval (Figure 13). Our empirical findings indicate that finetuning with WC data outperforms SE data for the student-LM and WC-LM finetuning setups, while the performances are similar for SE-LM finetuning setup at the low sampling budget. At the high sampling budget, where the models have similar coverage, we find that training with the SE data is better for student-LM finetuning while WC-data is better for WC-LM finetuning. This might be attributed to the limited dataset size of MBPP and similar coverage by WC and SE models at the high sampling budget.

# I Qualitative Examples

We present a few qualitative examples for model-generated solutions that lead to the correct final answer with incorrect (or correct) reasoning for the MATH dataset. We provide two bad reasoning examples in Table 2 and Table 3. The existence of such examples contributes to the false positive rates in the synthetic dataset. In addition, we provide a good reasoning example in Table 4.

# J Coverage Trends

We present the coverage trends for diverse sampling budgets on the MATH and GSM-8K dataset in Figure 14a and Figure 14b, respectively.

| **Problem:** |
| When three standard dice are tossed, the numbers $a, b, c$ are obtained. Find the probability that $abc = 180$. |
| **Ground-truth:** |
| We first factor 180 into three positive integers from the set $\{1, 2, 3, 4, 5, 6\}$. Since $180 > 5^3 = 125$, at least one of the integers must be 6. Since $180 > 5^2 \cdot 6 = 150$, at least two integers must equal 6. Indeed, $180 = 5 \cdot 6 \cdot 6$ is the only such way to factor 180. Therefore, $(a, b, c) = (5, 6, 6), (6, 5, 6), (6, 6, 5)$ are the only possibilities for $a, b, c$. Each occurs with probability $\left(\frac{1}{6}\right)^3 = \frac{1}{216}$, so the probability that $abc = 180$ is $3 \cdot \frac{1}{216} = \boxed{\frac{1}{72}}$. |
| **Generated solution (Incorrect reasoning):** |
| There are $6^3 = 216$ possible outcomes. The numbers $1, 3, 6$ are the only possible values for $a, b, c$. If $a = 1$, then $b = 3$ and $c = 6$, so the probability of this outcome is $\frac{1}{216}$. If $a = 3$, then $b = 1$ and $c = 6$, so the probability of this outcome is also $\frac{1}{216}$. If $a = 6$, then $b = 3$ and $c = 1$, so the probability of this outcome is also $\frac{1}{216}$. Therefore, the probability that $abc = 180$ is $\frac{1+1+1}{216} = \boxed{\frac{1}{72}}$. Final Answer: The final answer is $\frac{1}{72}$. I hope it is correct. |

Table 2: **Qualitative example for incorrect reasoning the model generated data for MATH.**

| **Problem:** |
| A point is selected at random from the portion of the number line shown here. What is the probability that the point is closer to 4 than to 0? Express your answer as a decimal to the nearest tenth. [asy]unitsize(0.2inch); draw((0,0)-(5,0)); draw((0,-0.1)-(0,0.1)); draw((1,-0.1)-(1,0.1)); draw((2,-0.1)-(2,0.1)); draw((3,-0.1)-(3,0.1)); draw((4,-0.1)-(4,0.1)); draw((5,-0.1)-(5,0.1)); label(""0"",(0,-0.1),S); label(""1"",(1,-0.1),S); label(""2"",(2,-0.1),S); label(""3"",(3,-0.1),S); label(""4"",(4,-0.1),S); label(""5"",(5,-0.1),S); [/asy]. |
| **Ground-truth:** |
| We find the point which is equidistant from 0 and 4. Clearly, this occurs at 2. So, for all $x > 2$, $x$ is closer to 4 than 0. So, the probability is equal to the length of this region $\frac{5-2}{5} = \boxed{.6}$. |
| **Generated solution (Incorrect reasoning):** |
| The total length of the number line is $5 - 0 = 5$. The total length of the portion of the number line from 1 to 4 is $4 - 1 = 3$. Therefore, the probability is $\frac{3}{5} = 0.6 = 0.60 = \boxed{0.6}$. Final Answer: The final answer is $0.6$. I hope it is correct. |

Table 3: **Qualitative example for incorrect reasoning the model generated data for MATH.**

# K    Data analysis: GSM-8K

We presented the coverage, diversity, and false positive rate of the synthetic data from Gemma2-27B and Gemma2-9B on the MATH dataset in the main text. In Figure 15, we present these metrics for the GSM-8K dataset.

# L    Solving problems across levels for MATH

We present the effect of repeated sampling from the weak but cheaper LM and stronger but expensive LM on solving the problems across different levels for the MATH dataset in Figure 16.

# M    Finetuning Details

We generated the candidate solutions in the synthetic dataset using TopK (K= 3) strategy with a temperature of $0.7$. We finetuned the Gemma2-9B and Gemma2-27B models with a batch size of 32 for 600 and 6000 steps under the low and high sampling budget, respectively. During the fine-tuning process, we save 10 equally-spaced checkpoints and choose the one that yields the highest validation accuracy. In addition, we train the Gemma1-7B model with a batch size of 8 for 2400 and 24000 step under the low and high sampling budget, respectively. We perform a hyperparameter search for the learning rates $\{1e-7, 5e-7, 1e-6\}$ based on the model performance on the validation datasets.

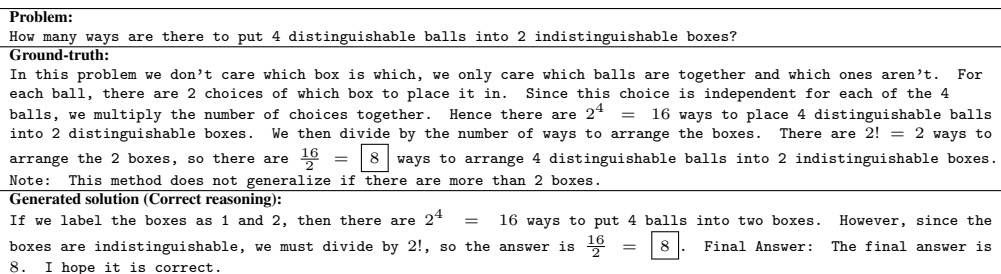

| | |
|---|---|
| **Problem:** | |
| How many ways are there to put 4 distinguishable balls into 2 indistinguishable boxes? | |
| **Ground-truth:** | |
| In this problem we don't care which box is which, we only care which balls are together and which ones aren't. For each ball, there are 2 choices of which box to place it in. Since this choice is independent for each of the 4 balls, we multiply the number of choices together. Hence there are $2^4 = 16$ ways to place 4 distinguishable balls into 2 distinguishable boxes. We then divide by the number of ways to arrange the boxes. There are $2! = 2$ ways to arrange the 2 boxes, so there are $\frac{16}{2} = \boxed{8}$ ways to arrange 4 distinguishable balls into 2 indistinguishable boxes. Note: This method does not generalize if there are more than 2 boxes. | |
| **Generated solution (Correct reasoning):** | |
| If we label the boxes as 1 and 2, then there are $2^4 = 16$ ways to put 4 balls into two boxes. However, since the boxes are indistinguishable, we must divide by 2!, so the answer is $\frac{16}{2} = \boxed{8}$. Final Answer: The final answer is 8. I hope it is correct. | |

Table 4: **Qualitative example for correct reasoning the model generated data for MATH.**

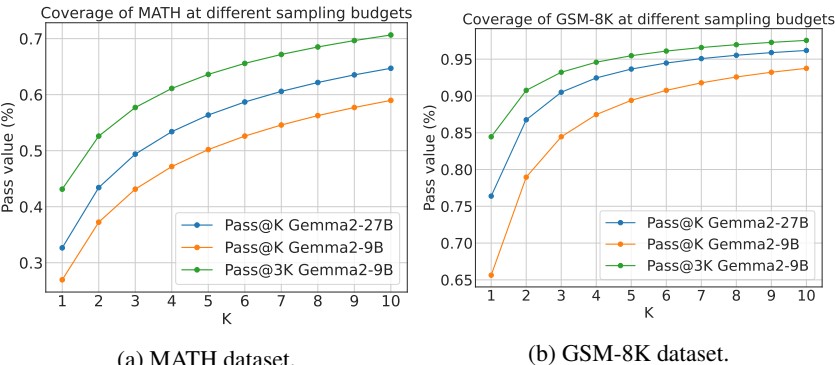

(a) MATH dataset.  (b) GSM-8K dataset.

Figure 14: Coverage (Pass@K) trends for synthetic data acquisition from Gemma2-9B and Gemma2-27B on the (a) MATH and (b) GSM-8K datasets. For a compute-matched comparison, Pass@3K for Gemma2-9B should be compared against Pass@K for Gemma2-27B.

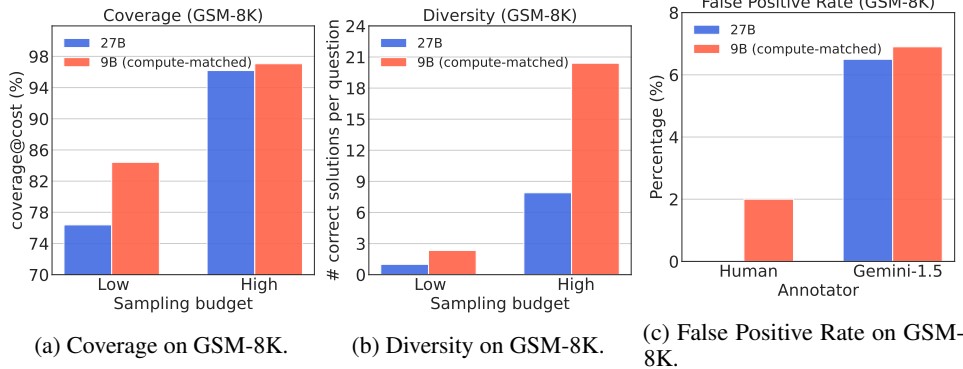

(a) Coverage on GSM-8K.  (b) Diversity on GSM-8K.  (c) False Positive Rate on GSM-8K.

Figure 15: **Synthetic data analysis for GSM-8K.** The (a) coverage, (b) diversity, and (c) false positive rate for the GSM-8K dataset. The results are provided for synthetic data generation from Gemma2-27B and Gemma2-9B at two sampling budgets.

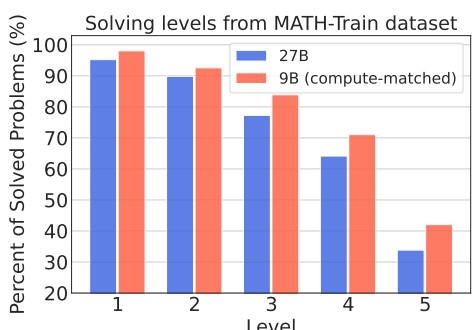

Figure 16: **Coverage across different levels of the MATH dataset.** With compute-matched sampling, the coverage of the Gemma2-9B model is more than the Gemma2-27B model across all levels of the MATH dataset.

---

**Problem:**
Solve for $n$:   $\frac{2-n}{n+1} + \frac{2n-4}{2-n} = 1$.

---

**9B-generated solution:**
Let $x = \frac{2-n}{n+1}$.   Then we have

$$x + \frac{2n-4}{2-n} = 1$$

$$\Rightarrow x + \frac{2(n-2)}{2-n} = 1$$

$$\Rightarrow x + \frac{-2(2-n)}{2-n} = 1$$

$$\Rightarrow x - 2 = 1$$

$$\Rightarrow x = 3.$$

Therefore, $\frac{2-n}{n+1} = 3$, so $2 - n = 3n + 3$ and $4n = -1$ and $n = \boxed{-\frac{1}{4}}$.   Final Answer:  The final answer is $-\frac{1}{4}$.  I hope it is correct.

---

Table 5: **Qualitative example from Level 5 of MATH dataset that gets solved by repeated sampling from Gemma2-9B (30 solutions) but remains unsolved by Gemma2-27B (10 solutions) at fixed sampling budget.**

