# OpenReview forum: "Smaller, Weaker, Yet Better: Training LLM Reasoners via Compute-Optimal Sampling"
_NeurIPS.cc/2024/Workshop/MATH-AI — MATH-AI 24_

### Official Review · Reviewer_Tp1X · 2024-10-03
**Good topic and well-written**

**Rating:** 8
**Confidence:** 3

**Review:**

The paper investigates whether using weaker, less compute-intensive language models (LMs) for generating synthetic training data is more effective than stronger, more expensive LMs. It evaluates the trade-offs in terms of coverage, diversity, and false positive rates between weaker but cheaper (WC) and stronger but expensive (SE) models, and compares their impact across several benchmarks. The results show that training models with WC-generated data consistently outperforms SE-generated data, challenging the common practice of relying on SE models.

---
## Strength:
- **Innovative Approach:** The paper introduces a novel "weak-to-strong" improvement method, where a weaker LM can teach a stronger LM, adding a new perspective to model training. This method achieve sota in various benchmark comparing to using stronger but expensive ones.
- **Comprehensive Evaluation:** The paper evaluates different setups (knowledge distillation, self-improvement, weak-to-strong improvement) using multiple models and benchmarks, including MATH and GSM-8K datasets.
- **Cost-Effective:** It presents a strong argument for the use of weaker models, which could make training more affordable and accessible by optimizing compute resources.

---
## Weakness:
- **Higher False Positive Rates:** Weaker models introduce higher false positive rates in the generated synthetic data, which could affect downstream tasks. It's indeed a compute-optimal method, but maybe the best method for synthetic high-quality data for distilling student models.

---
Overall, I think this paper is a well-written and novel submission for a workshop. Therefore, I recommend an accept.

---

### Official Review · Reviewer_4onj · 2024-10-06

**Rating:** 8
**Confidence:** 4

**Review:**

* Pros: the research problem proposed by this study is novel and interesting; the authors conduct experiments considering extensive tuning scenarios and metrics to measure the synthetic dataset.

* Cons: I suggest the author conduct the experiment with higher computational budgets, to see whether the SE model can finally beat the WC model.

---

### Official Review · Reviewer_A8Ra · 2024-10-08

**Rating:** 6
**Confidence:** 3

**Review:**

### Summary:
The paper addresses the trade-offs in synthetic data generation for large language models (LLMs) by comparing two approaches: using a stronger but more expensive model (SE) versus a weaker but cheaper model (WC) under a fixed computational budget. It explores how synthetic data from WC models may have higher coverage and diversity, though at the cost of higher false positive rates. The authors propose that using data from WC models can lead to more compute-efficient training, yielding better performance than SE models across multiple reasoning tasks, including knowledge distillation, self-improvement, and a novel weak-to-strong improvement setup. Experiments on benchmarks such as MATH and GSM-8K demonstrate the advantages of using WC-generated data in various finetuning paradigms.

---

### Strengths:
1. **Important Problem**: The paper tackles an essential problem in the field—optimizing the generation of synthetic data for improving LLM reasoning capabilities. This is highly relevant as training large models is often computationally expensive, and identifying more cost-effective approaches is beneficial.

2. **Insights from Results**: The results reveal key insights, showing that models trained on WC-generated data can outperform those trained on SE-generated data, contrary to the common practice of relying on stronger models. This insight challenges existing assumptions in the field and paves the way for more cost-efficient LLM training methodologies.

---

### Weaknesses:
1. **Limited Model Inclusion**: The paper only considers the *Gemma* series models in its experiments, missing the opportunity to include a wider range of models like the *LLaMa* series or other state-of-the-art LLMs. Including more diverse models would strengthen the generalizability of the findings and ensure that the proposed approach works across different LLM architectures.

2. **Generalizability Concerns**: The generalizability of the findings may be limited due to the narrow set of models tested. While the results are promising, a broader evaluation across various LLM families, such as *LLaMa* or *Mistral*, could provide more robust conclusions.

---

### Decision · Program_Chairs · 2024-10-09

Accept